# The probabilistic nature of dune collisions in 2D

Paul A. Jarvis[1,2,3], Clement Narteau[4], Olivier Rozier[4], and Nathalie M. Vriend[5,1,6,7]

[1]Department of Applied Mathematics and Theoretical Physics, University of Cambridge, Wilberforce Road, Cambridge, CB3 0WA, UK.
[2]Department of Earth Sciences, University of Geneva, Rue des Maraîchers 13, 1205, Geneva, Switzerland
[3]GNS Science, 1 Fairway Drive, Lower Hutt, 5011, New Zealand
[4]Institut de Physique du Globe de Paris, Sorbonne Paris Cité, Université Paris Diderot, 1 Rue Jussieu, Paris, 75005, France.
[5]BP Institute, University of Cambridge, Madingley Road, Cambridge, CB3 0EZ, UK.
[6]Department of Earth Sciences, University of Cambridge, Madingley Road, Cambridge, CB3 0EZ, UK.
[7]College of Engineering and Applied Science, University of Colorado Boulder, 1111 Engineering Drive, Boulder, Colorado, USA.

**Correspondence:** Paul A Jarvis (p.jarvis@gns.cri.nz)

**Abstract.**

Dunes are bedforms of different size and shape, appearing throughout aeolian, subaqueous and extra-terrestrial environments. Collisions between dunes drive dune field evolution, and are a direct result of interacting dunes of different heights, travelling at different speeds. We perform 2D cellular automaton simulations of collisions between dune pairs migrating in a steady flow. Modelled collisions can result in either ejection, where dunes exchange mass before separating, or downstream- or upstream-dominant coalescence (merging of dunes). For each of these three elementary types of interaction, we identify the mass exchange mechanism and the distinctive intermediate morphologies. Surprisingly, we show that the collision outcome depends probabilistically on the initial dune area ratio $r$ and can be described by a narrow sigmoidal function centred on $r = 1/2$. Finally, we compare our simulations with laboratory experiments of dune collisions, finding good agreement concerning the intermediate morphology and the collision outcome. Our results can motivate further observational or experimental studies that validate our probabilistic collision predictions and fully determine the controls on the coalescence-ejection transition.

## 1 Introduction

Dunes are self-organising structures that form spontaneously on a particle-laden surface overlain by a sufficiently strong flowing medium. In nature, they can be found in aeolian landscapes, e.g., deserts (Elbelrhiti et al., 2008; Lü et al., 2021) or coastal beaches (Parteli et al., 2006), aqueous environments such as river beds (de Almeida et al., 2016) or subjected to extraterrestrial atmospheres (Bishop, 2007). In these settings, dunes have long been thought to grow to attain a maximum height which, in settings with sufficient sediment, could be controlled by the flow depth (in aqueous environments) or the thickness of the planetary boundary layer (for aeolian systems) (Andreotti et al., 2009; Andreotti and Claudin, 2013). Alternatively, sediment supply may be an additional limiting factor (Gunn et al., 2022; Jarvis et al., 2022). Regardless, the migration velocity $c$ of a dune depends inversely on the dune size. Based on mass conservation, it has frequently been proposed that $c \sim 1/\mathcal{H}$ (Bagnold,

1941; Southard, 1991), where $\mathcal{H}$ is the dune height, but other relations, including $c \propto 1/\mathcal{L}$ (Kroy et al., 2002), where $\mathcal{L}$ is the dune length, or $c \propto 1/(\mathcal{H} + \mathcal{H}_0)$ (Andreotti et al., 2002a), where $\mathcal{H}_0$ is a constant, have also been suggested.

The inverse relationship between dune size and velocity means that faster, smaller dunes can approach and collide with slower, larger dunes. Such collisions have been frequently observed in subaqueous experiments (Coleman and Melville, 1994; Bradley and Venditti, 2019; Jarvis et al., 2022) and numerical simulations (Gao et al., 2015), as well as for aeolian dunes (Lü et al., 2021) even though the latter evolve over much longer timescales. These collisions have been shown to play an important role in pattern coarsening, whereby a larger number of smaller dunes transition to become a smaller number of larger dunes (Coleman and Melville, 1994; Gao et al., 2015; Bradley and Venditti, 2019; Jarvis et al., 2022). This coarsening process occurs during the development of a dune field from a flat bed whereas more mature dune fields can attain a steady state, where their mean wavelength and amplitude remain relatively constant. In 2D, coarsening primarily occurs through dune coalescence, where a pair of colliding dunes merge. Two distinct types of coalescence have been observed, downstream- and upstream-dominant, defined according to which peak survives the coalescence process (Coleman and Melville, 1994; Gao et al., 2015; Jarvis et al., 2022). Another possible collision outcome that has been recognized experimentally is ejection, where mass is transferred from the downstream larger, slower dune, to the upstream, smaller, faster dune, until the leading dune becomes sufficiently small that it can accelerate away (Diniega et al., 2010; Gao et al., 2015; Jarvis et al., 2022). The balance between the different outcomes ultimately means that dune collisions are a strong and dominant control on dune field evolution (Elbelrhiti et al., 2008; Durán et al., 2009; Kocurek et al., 2010; Hugenholtz and Barchyn, 2012), regulating both the size and the spacing of dunes (Hersen and Douady, 2005; Génois et al., 2013a, b).

Although this current study only considers the collision of 2D dunes, for which only coalescence and ejection have been observed, it is important to acknowledge that collisions between 3D barchan dunes can produce a much wider range of outcomes owing to turbulence and lateral sediment transport (Endo et al., 2004; Durán et al., 2005; Elbelrhiti et al., 2005; Hersen, 2005; Katsuki et al., 2011; Assis and Franklin, 2020, 2021). Additionally, it has also been shown that a turbulent wake shed by an upstream dune (Bristow et al., 2018, 2019, 2020) can enhance the migration velocity of a downstream neighbour or even cause it to split into two (Elbelrhiti et al., 2005; Worman et al., 2013), further complicating collision dynamics (Bacik et al., 2020, 2021; Assis and Franklin, 2021). Although recirculation zones downstream of dune crests occur in both 2D and 3D configurations (Hermann et al., 2005; Araùjo et al., 2013; Michelsen et al., 2015), turbulence itself is an inherently 3D phenomenon, even present in quasi-2D experiments (Bacik et al., 2020, 2021). Thus, in a pure 2D domain and for dunes sufficiently large to be scale invariant (Kroy et al., 2002; Andreotti et al., 2002b), only the size ratio between dunes controls the collision outcome. Katsuki et al. (2005) developed a simplified model of 2D dune collision and showed that for $r = A_u/A_d \gtrsim 0.5$, where $A_u$ and $A_d$ are the areas of the upstream and downstream dunes, respectively, ejection should occur. Conversely, coalescence is predicted for $r \lesssim 0.5$. A more sophisticated continuum model from Diniega et al. (2010) which accounts for the dependence of sediment flux on shear stress, however, predicted the transition to occur for $r \approx 1/3$. For 3D barchans, both Katsuki et al. (2011) and Bo and Zheng (2013) found the transition also depended on the perpendicular distance between the dune axes (axis offset distance). Later, Zhou et al. (2019), motivated by results of Large Eddy Simulations of on-axis collisions between 3D barchans, suggested ejection occurs if the sand flux received by the downstream dune exceeds what is lost from the barchan

horns. These predictions and hypotheses remain largely unverified, either through field observations or laboratory experiments. However, Assis and Franklin (2020) showed experimentally that, for subaqueous 3D barchans, all collision outcomes can be mapped in a regime space defined by the Shields number (proxy for flow strength), the axis offset distance and the dune size ratio. Such a regime diagram has not been experimentally created in a 2D system though.

In this paper, we use the cellular automaton model ReSCAL (Narteau et al., 2009; Zhang et al., 2010; Rozier and Narteau, 2013) to simulate the collision of 2D dunes and quantitatively constrain the coalescence-ejection transition. We are able to produce both downstream- and upstream-dominant coalescence outcomes, as well as instances of ejection. Such interactions have previously been generated in ReSCAL simulations of bedform coarsening (Gao et al., 2015), although this study represents a first attempt to constrain the transition between regimes. Limiting our domain to 2D results in a narrower range of outcomes

than can be observed in natural 3D systems. However, this means we can more completely study the phenomena, something that would be computationally very expensive in the much larger parameter space of 3D systems. By performing large numbers (1600) of simulations for dune pairs with the same size ratio, we show that the collision outcome can be modelled probabilistically rather than deterministically, and find an empirical relationship for the coalescence-ejection transition. Additionally, we note that intermediate structures during collisions are associated with distinct morphologies. Finally, we compare these results

with qualitative and quantitative observations of colliding dunes in subaqueous experiments reported by Jarvis et al. (2022). Although our study is strictly only valid for 2D systems, our results should motivate further research to test if the fundamental characteristics of collisions that we observe can be preserved in 3D environments, where turbulence and flow perturbations, as well as lateral sediment transport during interactions, also have an influence.

## 2    Methods

We simulate interactions between discrete 2D dunes using the dune model ReSCAL (Narteau et al., 2009) which couples a cellular automaton model of sediment transport and a lattice gas model of turbulent fluid flow. We summarise the numerical method below whilst full details can be found in Narteau et al. (2009) and Rozier and Narteau (2013).

Simulations consider a two-dimensional domain of square cells of length $l_0$ in one of the following states: fluid, neutral, immobile sediment, mobile sediment. Neutral cells denote the upper and lower boundaries. Sediment transport is modelled

using transitions of pairs of nearest-neighbour cells corresponding to physical processes (erosion, deposition, transport). Additionally, a repose angle of the sediment is imposed to account for avalanching. Whilst direct numerical simulations (Lefebvre and Winter, 2016) and experiments (Kwoll et al., 2016) on subaqueous dunes have shown that the leeside slope angle strongly controls the flow downstream of the dune, with low-angle dunes significantly reducing turbulence and flow separation, we fix the angle of repose at $35°$. We briefly discuss the implications of this choice in Section 4. Each process has a characteristic

timescale expressed in units of $t_0$ (the model time-step), which determines the probability of a particular transition to occur. This probabilistic approach means the model is not entirely deterministic, but takes a random seed as an input.

The fluid flow is simulated using a lattice gas model. Particles move according to their velocity vectors that can change according to collisions. Particle fluxes are averaged to produce a velocity field. Particles colliding with the upper boundary

conserve their horizontal velocity but experience a change of sign to their vertical velocity, creating a free-slip boundary
condition. At the lower boundary, all fluid particles colliding with the surface rebound in their incident direction, creating a
no-slip condition. This reproduces the expected logarithmic velocity profile for turbulent flow over a flat wall (Narteau et al.,
2009), with surface shear stress $\tau_s$ defined as the normal derivative of the velocity field with respect to topography. The erosion
rate is defined for three distinct regions as

$$
\Lambda_e = \begin{cases}
0, & \text{for } \tau_s < \tau_1, \\
\Lambda_0(\tau_s - \tau_1)/(\tau_2 - \tau_1), & \text{for } \tau_1 \leq \tau_s \leq \tau_2, \\
\Lambda_0, & \text{for } \tau_s > \tau_2,
\end{cases}
\tag{1}
$$

where $\Lambda_0$ is a constant, $\tau_1$ the critical shear stress for sediment motion and $(\tau_2 - \tau_1)^{-1}$ the linear coefficient between $\Lambda_e$ and
$\tau$ ($\tau_2$ is an adjustable parameter). This erosion, along with transport and deposition, modifies the lower boundary of the fluid
domain, providing feedback on the fluid flow. This two-way coupling generates acceleration of the flow on the upstream side
of dunes, shear layers and a recirculation zone on the downstream side (Narteau et al., 2009; Zhang et al., 2010), as observed
naturally (Sweet and Kocurek, 1990) or imposed by other numerical models (Hermann et al., 2005).

   We use a two-dimensional periodic domain of height $H = 300\,l_0$ and length $L = 2000\,l_0$ (see Appendix A for justification
that the domain is sufficiently large). Two triangular sand piles with slope angle $35°$ (consistent with the angle of repose used
in previous cellular automaton simulations, e.g., Zhang et al. (2014)) were placed $1000\,l_0$ apart. The slope angle was equal
to the repose angle. In all simulations, the downstream dune was initially larger than that upstream, ensuring interaction. We
set $\tau_1 = 0$ and $\tau_2 - \tau_1 = 1000\,\tau_0$ but found that the observed interactions did not depend on the absolute values for $\tau_2 \gg \tau_1$,
i.e., as long as the flow regime is far from the sediment transport threshold. This is to be expected since we are assuming
flow conditions are far above the threshold for sediment transport. Table B1 lists the parameter values used for the suite of
simulations.

## 2.1  Physical scaling

Later in this manuscript (Section 4), we will compare our simulated collisions with those observed in the subaqueous exper-
iments of Jarvis et al. (2022). However, since cellular automatons are defined on a discrete domain, and given the arbitrary
nature of the transition rules, there is no prior relationship between $l_0$ and the time-step $t_0$ and physical length and time scales.
Instead, we define these by comparing model results with experimental and natural observations, following Narteau et al.
(2009) and Zhang et al. (2014). For $l_0$, this was previously achieved (Narteau et al., 2009; Zhang et al., 2014) by simulating
fluid flow over an initially-flat sediment bed and determining the most unstable initial wavelength $\lambda_{max}$, which is predicted to
be $\lambda_{max} = 50(\rho_p - \rho_f)d/\rho_f$ (Hersen et al., 2002; Elbelrhiti et al., 2005; Andreotti and Claudin, 2013), where $\rho_{p(f)}$ is the sediment
(fluid) density and $d$ the particle diameter. Within ReSCAL, $\lambda_{max} \approx 40\,l_0$ (Narteau et al., 2009; Zhang et al., 2014) whilst in the
experiments of Jarvis et al. (2022), $\rho_p = 2500 \text{ kg m}^{-3}$, $\rho_f = 1000 \text{ kg m}^{-3}$ and $d = 1.21 \times 10^{-3}$ m resulting in $l_0 = 2.27 \times 10^{-3}$

m. We then determine the time-scale $t_0$ by comparing the modelled equilibrium sediment flux with field observations (Narteau et al., 2009; Zhang et al., 2014). For the saturated flux, we take (Meyer-Peter and Müller, 1948)

$$Q_{\text{sat}} = \frac{8\rho_{\text{f}}(u_*^2 - u_{*\text{t}}^2)^{3/2}}{(\rho_{\text{p}} - \rho_{\text{f}})g}. \tag{2}$$

We note that many similar transport laws exist for bedload transport in turbulent flow (Paintal, 1971; Bagnold, 1973; Engelund and Fredsoe, 1976; Fernandez-Luque and Beek, 1976; Bridge and Dominic, 1984; Lajeunesse et al., 2010; Pähtz and Durán, 2020), whose mathematical form only vary slightly. Since there is little experimental evidence distinguishing between these transport laws (Lajeunesse et al., 2010), we choose for simplicity equation 2. We also take (Bagnold, 1936; Iverson and Rasmussen, 1999)

$$u_{*\text{t}} = \frac{1}{10}\sqrt{\frac{(\rho_{\text{p}} - \rho_{\text{f}})gd}{\rho_{\text{f}}}} \tag{3}$$

for the threshold friction velocity. Substituting equation 3 into 2 gives

$$Q_{\text{sat}} = \frac{8\rho_{\text{f}}}{(\rho_{\text{p}} - \rho_{\text{f}})g}\left(u_*^2 - \frac{(\rho_{\text{p}} - \rho_{\text{f}})gd}{100\rho_{\text{f}}}\right)^{3/2}. \tag{4}$$

In the experiments of Jarvis et al. (2022), $u_*$ varies so we take an intermediate value of $0.1$ m s$^{-1}$. Equation 4 then gives $Q_{\text{sat}}$ = $5.29 \times 10^{-4}$ m$^2$ s$^{-1}$ which is matched with the modelled saturated flux (Narteau et al., 2009; Zhang et al., 2014) to determine that $t_0 = 9.74 \times 10^{-3}$ s. The computed values of $l_0$ and $t_0$ enable comparisons between our simulations and the experiments of Jarvis et al. (2022).

## 3 Results and discussion

### 3.1 Three elementary types of dune interaction

The simulations reproduce both coalescence and ejection behaviour, as shown in Fig. 1, where the initially-upstream dune has been coloured red, and the initially-downstream dune blue. Two distinct types of coalescence (Coleman and Melville, 1994; Gao et al., 2015), downstream- (Fig. 1a) and upstream-dominant (Fig. 1b) coalescence, feature distinctly different intermediate mechanics despite both resulting in only a single dune surviving the interaction. Figure 2 shows more details regarding the evolution of the positions and heights of the peaks and troughs in the intermediate structures.

Downstream-dominant coalescence (Fig. 1a) prevails if the upstream dune is much smaller than the downstream dune ($r = A_{\text{u}}/A_{\text{d}} \ll 1$). The upstream dune climbs the stoss-side of the downstream dune (raising the trough) but the increased shear stress on the stoss-slope of the larger dune spreads the upstream material, some of which is transported over the downstream crest and deposited on the lee-side. The trough increases in height faster than the upstream peak until they reach the same

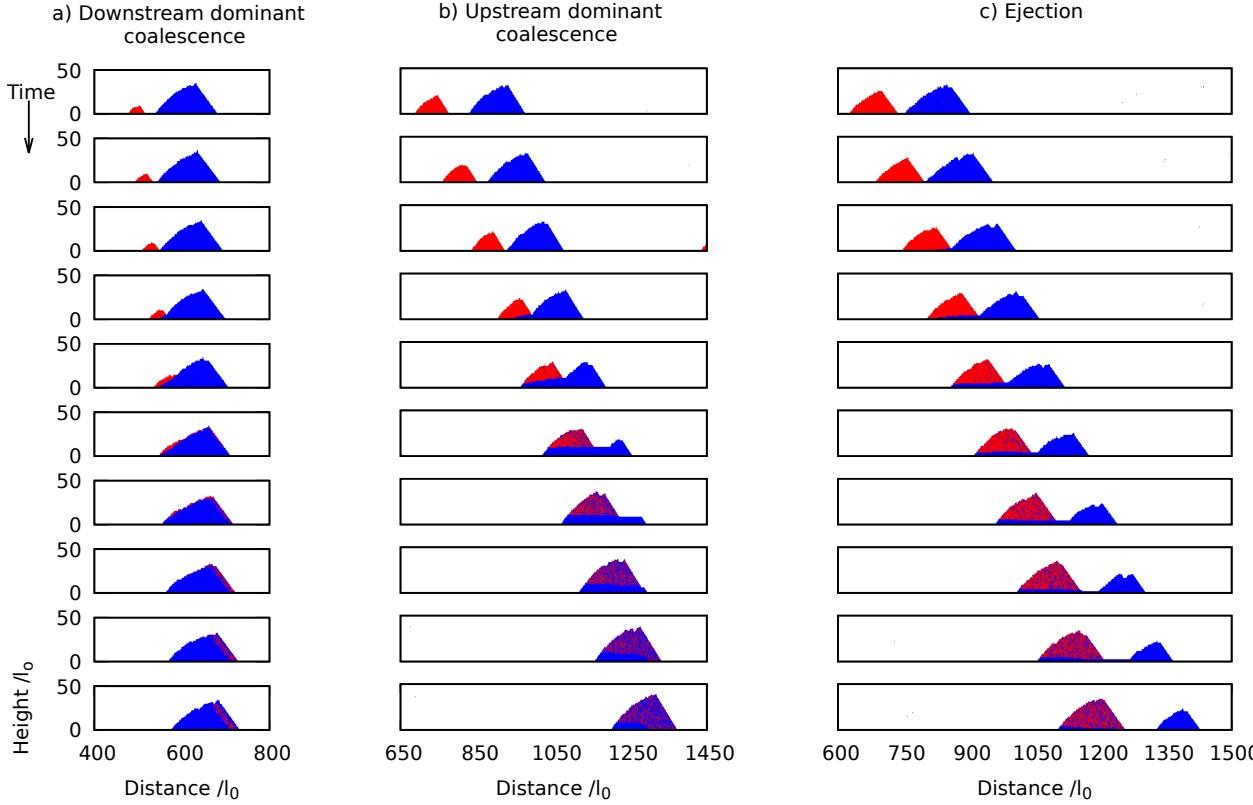

**Figure 1.** The three elementary types of dune interaction observed in the simulations: a) downstream-dominant ($A_u = 300$, $A_d = 3000$, $r = 0.1$), b) upstream-dominant ($A_u = 1200$, $A_d = 3000$, $r = 0.3$), and c) ejection ($A_u = 1800$, $A_d = 3000$, $r = 0.6$).

height and the upstream peak vanishes. The sediment which initially constituted the upstream dune (red in Fig. 1) ends up in a layer parallel to the slip face of the resultant dune. Migrating dunes typically contain many slip face-parallel layers (Bagnold, 1941; Allen, 1970) and, since the upstream material becomes incorporated in such a layer, evidence of the interaction is lost over time.

Upstream-dominant coalescence (Fig. 1b) preserves the peak of the upstream dune and occurs for larger values of ratio $r$ than downstream-dominant events. Upon touching, the upstream peak starts to climb the stoss-side of the downstream dune while retaining its avalanche face and recirculation zone. Sediment at the foot of the stoss-slope is trapped by the upstream dune and the gradient in shear stress between the new trough and the downstream peak increases. This means the downstream dune shrinks as it erodes faster, whilst the upstream dune grows due to the extra mass it has assimilated during its migration as a superimposed bedform. Meanwhile, the trough height increases until a flat plateau forms as the now smaller downstream peak accelerates away (Figs. 2b,e). The critical characteristic is that the downstream peak shrinks faster than it migrates; it is unable to escape and it sinks into the plateau which then disappears as the upstream peak migrates forward.

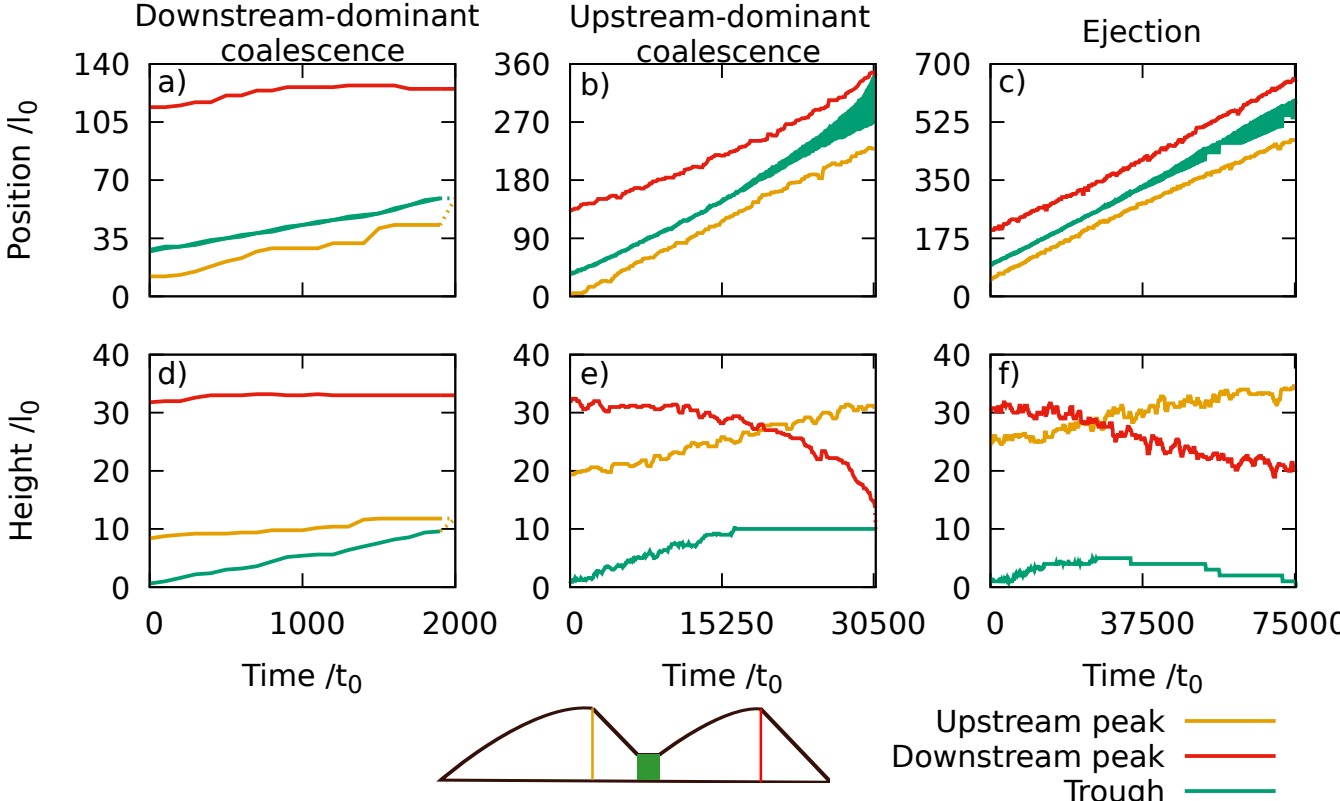

**Figure 2.** Peak and trough positions and heights in the intermediate bedform during a,d) downstream-dominant and b,e) upstream-dominant coalescence and c,f) ejection. $t = 0$ is defined as when the dunes first touch. The trough is defined as the range of points which form the miniumum height in the intermediate bedform so has finite extent (see sketch). In a,d), downstream-dominant coalescence completes once the trough and upstream peak coincide and is represented by dotted line segments.

The defining distinction between downstream- and upstream-dominant coalescence is the formation of a bounding surface within the internal structure of the merged dune, which extends into an open plateau located between two avalanche slopes. However, it is complicated to quantify the transition between the behaviours. Firstly, the simulated outcome is probabilistic, as will be demonstrated in the following section, so many simulations would be required to gain meaningful outcomes. Ad-
ditionally, plateaus always form for all superimposed bedforms reaching the crest (Text S1, Supporting Information), even though they may be very small and short-lived. This means that, close to the transition, superimposed bedforms on the dune surface make it challenging to track the separate peaks and, consequently, difficult to distinguish the two types of behaviour. Following the coalescence, the upper part of the resultant dune is well mixed with sediment from both the upstream and downstream dunes, whilst the lower part contains only initially-downstream material. The contact surface between the two
sediment-sources transitions from the original slope of the impacted dune to a horizontal plane. However, the bounding surface is ultimately eradicated as the dune migrates forward and material is transported from stoss to lee side.

For still larger values of $r$, ejection (Fig. 1c) occurs. As with upstream-dominant coalescence, the upstream peak grows at the expense of the downstream peak and the trough becomes a plateau. However, this time the plateau is continually eroded and shrinks, providing sediment to the downstream peak, slowing it's decrease in height. Eventually, the plateau in the trough is eroded, the upstream dune disconnects and the two peaks become distinct bedforms (Figs. 2c,f). After the interaction, the downstream dune still only contains initially-downstream sediment, whilst the upstream dune has a structure similar to that seen at the end of upstream-dominant coalescence. Again, this intermediate structure is ultimately lost. It needs noting that, close to the transition to coalescence, the plateau between the two peaks can be completely eroded in multiple locations, resulting in the ejection of multiple small bedforms. Nevertheless, since the interaction mechanism is the same regardless of whether one or more downstream dunes are ejected, we classify all of these events the same.

The exchange of mass resulting from the three different types of interaction produces distinctive internal structures within the resultant dunes, as can be seen in Fig. 1. For downstream-dominant coalescence, the upstream dune breaks-up as it climbs the stoss slope of the downstream-dune, with the associated sediment transport up to and over the crest. Avalanching and deposition on the lee-side creates a mixed layer, inclined at the angle of repose, within which the sediment from the initially-upstream dune is confined. Assuming there is no difference between the sediment in the two initial dunes, this layer will just appear as one of many that form in the final dune as it propagates (Bagnold, 1941; Allen, 1970) and, thus, no evidence of the interaction will be preserved. Conversely, in the case of upstream-dominant coalescence, it is the downstream dune which is lost. In this case, whilst some of the downstream material becomes mixed into the upstream dune as it propagates over the downstream stoss slope, much of the material becomes contained within a basal horizontal layer. This is preserved only for the time it takes for the final dune to propagate a distance equal to its length. During this time, the material from the basal layer is eroded at the foot of the stoss slope, and transported up and over the dune, resulting in the final dune becoming well-mixed. Finally, in the ejection case, whilst the downstream dune only loses sediment, the upstream dune initially gains material in a similar fashion to the upstream-dominant coalescence scenario. A basal layer of initially-downstream material forms in an entirely analogous way before ultimately becoming mixed into the upstream dune. Thus, for all three scenarios, although the collision creates distinct sedimentary structures, these are lost in the time it takes each dune to migrate a distance equal to its length. Therefore, these surfaces are unlikely to be preserved in the geological record except in zones of high deposition rates where dune interactions are synchronous with sediment accumulation.

### 3.2 Coalescence or ejection: Randomness in the interaction outcome

All transitions in the cellular automaton are governed by stochastic processes, as the model is taking a random seed as an input (Narteau et al., 2009). We find that the transition from coalescence to ejection can depend on the seed, i.e., the phenomenon can be described probabilistically as opposed to deterministically. Figures 3a-c demonstrate this variability for a case close to the transition, showing three different outcomes (respectively coalescence and ejection with two and one ejected dunes) for simulations with $A_{\mathrm{u}} = 2100\,l_0^2$ and $A_{\mathrm{d}} = 4000\,l_0^2$ ($r = 0.525$) but different seeds. Interactions that lead to an increase in the number of dunes only occur for values of $r$ close to the transition.

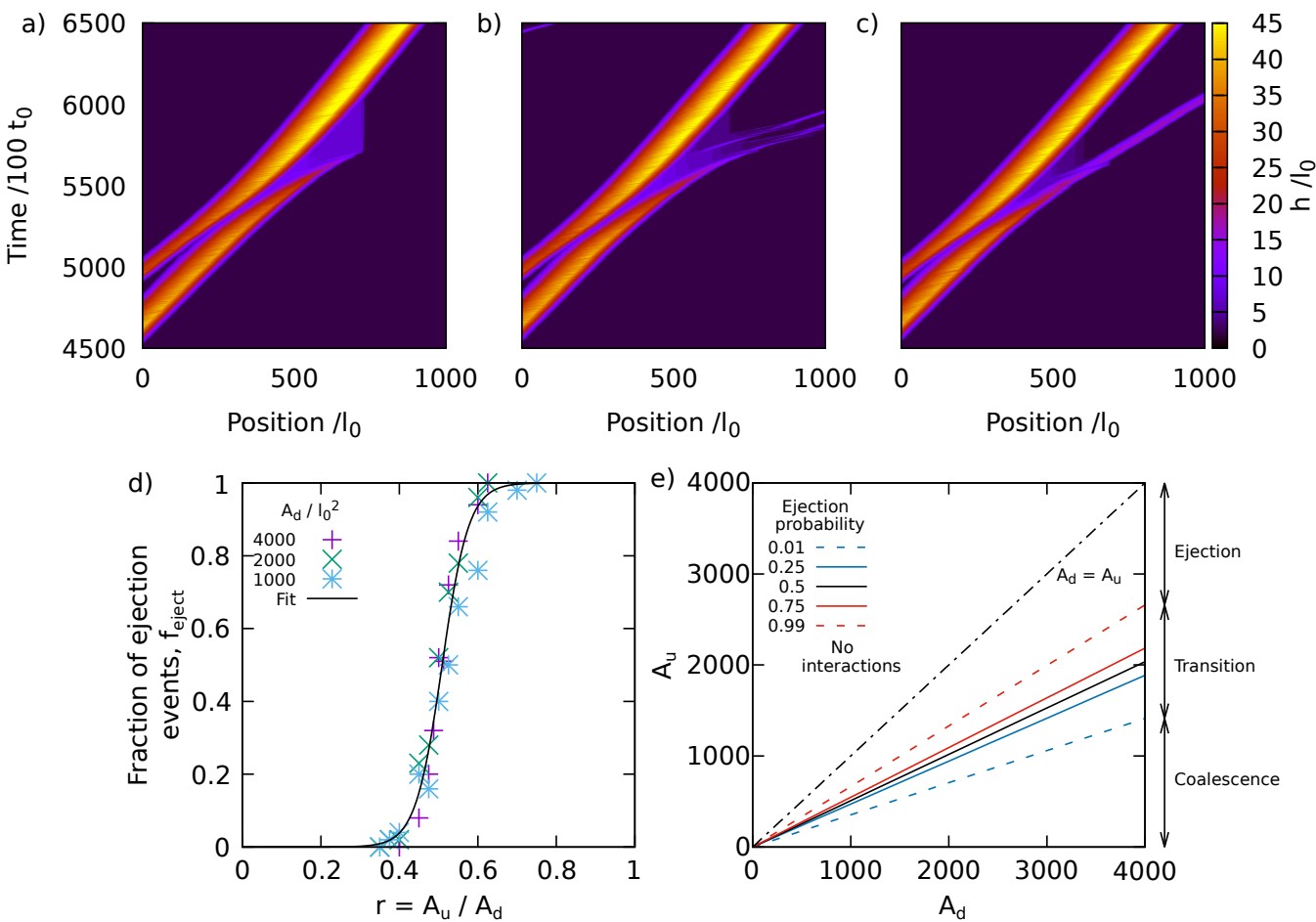

**Figure 3.** a,b,c) Interactions for $A_u = 2100\ l_0^2$ and $A_d = 4000\ l_0^2$ ($r = 0.525$) but different random seeds. The colour shows the height $h$ and time runs vertically. In a) coalescence occurs whilst in b) and c) ejection is observed, with b) showing multiple ejected bedforms. d) Fraction of coalescence events as a function of $r = A_u/A_d$, for different values of $A_d$. Variation in the data is due to only 50 simulations being used to produce each point. The curve is the fitted function $f = \{\tanh[a(r-b)]+1\}/2$, with $a = 16 \pm 3$ and $b = 0.506 \pm 0.006$. e) Contours of probability of ejection as a function of $A_u$ and $A_d$.

To estimate the outcome probability for a given $r$, we performed 50 simulations with different seeds for 32 different pairs of dune areas, resulting in a total of 1600 simulations. Performing further simulations would be prohibitively computationally expensive. Then, for each set, we quantified the proportion of simulations resulting in either outcome. Figure 3d shows the fraction of ejection events $f_{eject}$ as a function of $r$. We observe a narrow transition from no ejection events to all ejection events around $r = 1/2$, with no apparent dependency on the absolute dune size. We note that this transition value is identical to that found by Katsuki et al. (2005) but greater than that found by Diniega et al. (2010). Unfortunately, we have no theoretical explanation for this result. Nonetheless, we fit the data to

$$f_{\text{eject}} = \frac{\tanh[a(r-b)]+1}{2},\qquad(5)$$

finding $a = 14 \pm 2$ and $b = 0.509 \pm 0.005$. Although any sigmoidal function could be used, the hyperbolic tangent provides the best fitting according to a sum of squared residuals criterion. We use equation 5 to define contours of probability for interaction outcome in the space $\{A_{\text{u}}, A_{\text{d}}\}$ (Fig. 3e). It's important to note that the precise fitted values of $a$ and $b$, and therefore the precise locations of the regime domains in Figure 3e, are strictly only valid in 2D systems.

## 4  Comparison between dune interactions in the experiments and the simulations

We compare both the qualitative morphology of interacting bedforms in the experiments of Jarvis et al. (2022) and the simulations, as well as the quantitative regime transition for ejection and coalescence. The experiments involved the creation and evolution of discrete, quasi-2D dunes from a thin layer of glass beads in a narrow, counter-rotating, water-filled annular flume. Full details can be found in Jarvis et al. (2022) but, during the experiment, dunes were observed to undergo both coalescence and ejection interactions with their neighbours. Given the experiments enabled measurements of the heights of the dunes undergoing interactions, we are able to compare the observed behaviour with the simulation results presented here. However, there are caveats to consider. Firstly, we only simulated interactions between two discrete dunes whilst, in the experiments, there is a train of interacting dunes. Secondly, the number of simulations (50) for each pair of dune areas is relatively small. However, since we have performed simulations for 28 different pairs of dune areas, we expect further simulations to just reduce the uncertainties on $a$ and $b$. Thirdly, dunes in the experiments have finite width equal to that of the channel (9 cm) and some three-dimensional effects may impact the results (dunes are typically longer at the outer wall than the inner wall by $\sim 6\%$). In particular, turbulent eddies will be shed by the dunes despite the narrow channel. Such 3D turbulent flow fields and their effect on dune collision dynamics cannot be reproduced in our 2D model. Finally, we observe experimental leeside slope angles $\theta \approx (18 \pm 2)°$, whilst in our simulations we set $\theta = 35°$. Although we have performed some additional simulations to verify that ejection and coalescence occur for $\theta = 18°$ as they do for $\theta = 35°$, we currently negelect any influence $\theta$ may have on the ejection-coalescence transition. Given these caveats, we identified experimental interactions (see Table 1) for comparison according to specific criteria: 1) events took place between discrete dunes and 2) during the interaction there was no physical overlap with other neighbouring dunes. Neverless, it is necessary to state that the existence of a dune train will cause flow disturbances that could still affect collision outcomes. These events were then labelled as coalescence or ejection. Jarvis et al. (2022) reported that downstream-dominant coalescence was not observed in any experiment as the majority of bedforms were of comparable size due to the initial conditions. This is consistent with the prediction from the numerical results, which show that a large initial size difference is required to feature downstream-dominant coalescence. Jarvis et al. (2022) also did not report any observations of dune repulsion or collision suppression, as seen by Bacik et al. (2020) in their experiments on interactions between isolated dune pairs in periodic domains. This is likely because dune repulsion acts to push a two-dune system towards an antipodal configuration (Bacik et al., 2021). However, at any given time, the dunes in the experiments of Jarvis

et al. (2022) were always relatively evenly spaced. Consequently, any dune repulsion effects would have been very small and difficult to observe.

**Table 1.** Upstream and downstream heights, $H_u$ and $H_d$, respectively, along with the corresponding area ratio $r$ and collision outcomes of the selected experiments from Jarvis et al. (2022). Also shown is the predicted probability of ejection $f_{eject}$ as given by equation 5. Where $f_{eject} > 0.99$, equation 5 gives a probability of ejection that, within uncertainty, is equal to 1.

|  | $H_u$ /mm | $H_d$ /mm | $r$ | Outcome | $f_{eject}$ |
|---|---|---|---|---|---|
| 1 | 9.7 | 26.4 | 0.12 | Coalescence | $1.9 \times 10^{-5}$ |
| 2 | 15.9 | 42.8 | 0.12 | Coalescence | $2.0 \times 10^{-5}$ |
| 3 | 15.6 | 30.2 | 0.25 | Coalescence | $7.1 \times 10^{-4}$ |
| 4 | 26.0 | 46.3 | 0.30 | Ejection | $2.8 \times 10^{-3}$ |
| 5 | 18.3 | 31.5 | 0.32 | Coalescence | $5.2 \times 10^{-3}$ |
| 6 | 18.8 | 30.4 | 0.37 | Coalescence | $1.9 \times 10^{-2}$ |
| 7 | 16.9 | 27.4 | 0.37 | Coalescence | $2.0 \times 10^{-2}$ |
| 8 | 17.5 | 28.2 | 0.37 | Coalescence | $2.2 \times 10^{-2}$ |
| 9 | 21.5 | 34.0 | 0.39 | Coalescence | $3.2 \times 10^{-2}$ |
| 10 | 23.2 | 34.5 | 0.44 | Ejection | 0.13 |
| 11 | 27.5 | 40.6 | 0.45 | Ejection | 0.16 |
| 12 | 36.5 | 49.7 | 0.54 | Ejection | 0.68 |
| 13 | 32.4 | 43.6 | 0.55 | Coalescence | 0.75 |
| 14 | 38.0 | 49.3 | 0.59 | Ejection | 0.92 |
| 15 | 26.9 | 32.6 | 0.69 | Ejection | 0.99 |
| 16 | 28.5 | 34.2 | 0.70 | Ejection | > 0.99 |
| 17 | 24.9 | 27.6 | 0.83 | Ejection | > 0.99 |
| 18 | 39.0 | 40.7 | 0.95 | Ejection | > 0.99 |
| 19 | 28.2 | 28.4 | 0.99 | Ejection | > 0.99 |

Morphologically, interacting bedforms in the experiments and simulations are similar. A distinctive feature of upstream-dominant coalescence seen in both datasets is the formation of a plateau towards the end of the interaction. Figures 4a and b compare the morphology of an experimental and a numerical example, respectively. The plateau is seen in both cases and was ubiquitous in all upstream-dominant coalescence events. The same intermediate morphology was observed experimentally by Groh et al. (2009) and in active dune fields (Gao et al., 2015). This plateau could have implications for boundaries in the

sedimentary record of previous dune-forming environments.

    Figure 4c shows the location of the selected events in the regime diagram (as determined from simulations), which has been transformed to be in terms of dune heights as opposed to areas, since height is easier to measure experimentally. Given the assumption of dune self-similarity (Diniega et al., 2010), the height and area measurements will be equivalent. There is

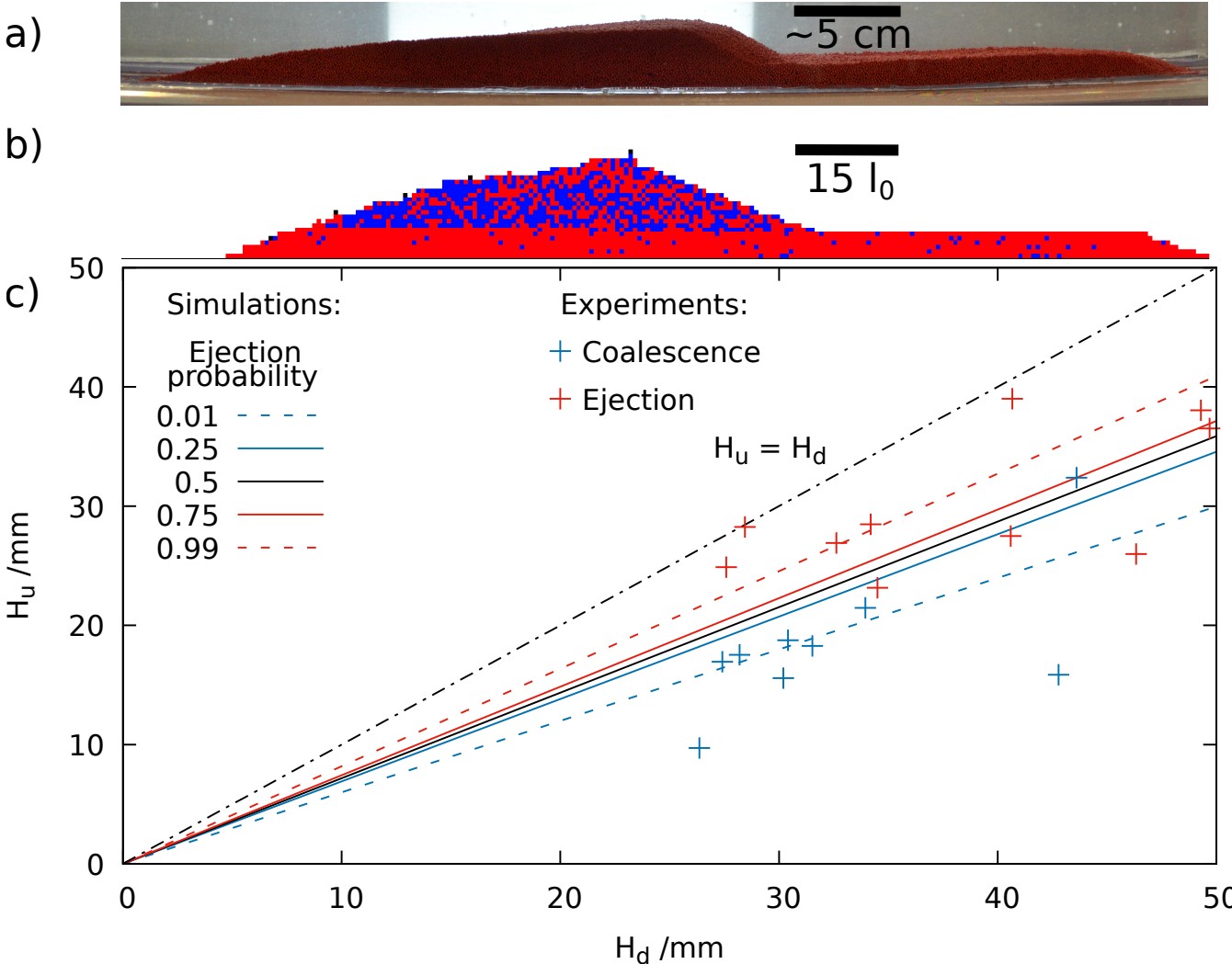

**Figure 4.** The morphology of intermediate bedforms in a) experiments of Jarvis et al. (2022) and b) simulations. Both show the distinctive plateau. c) The location of experimental ejection and coalescence events on the numerically-determined regime diagram. Most interactions result in their most likely outcome, with just four having a probability less than 0.5.

reasonable agreement between experiments and numerical predictions, with most interactions resulting in their most likely

outcome; only one coalescense event in nine and three ejection events in ten had a probability less than 0.5. One ejection event, though, only had a probability of 0.002 according to our empirical model, suggesting that there may be other factors at play. This is unsurprising since in the experiments, the dunes interact with multiple neighbours, whilst the simulations consider only a pair of dunes. This can lead to changes in the sand flux entering and leaving an interacting pair of dunes which may affect the outcome. Appendix C presents a visual means of comparing the experimental observations with the empirical probability

distribution determined by the simulations.

## 5 Concluding remarks

Dune collisions are an important part of dune field evolution and pattern coarsening and, in 2D, can result in two possible outcomes: coalescence and ejection (Coleman and Melville, 1994; Endo et al., 2004; Durán et al., 2005; Katsuki et al., 2005; Diniega et al., 2010; Bo and Zheng, 2013; Gao et al., 2015). Here we use a cellular automaton model (Narteau et al., 2009) to simulate collisions between discrete two-dimensional dunes and show that the collision outcome can be modelled as probabilistically depending on the dune size ratio. The observation that this is a probabilistic, rather than a deterministic, dependence is an interesting result which, to the authors' knowledge, has not been made before. Whilst this requires experimental validation, it does not seem unreasonable given that sediment transport is, by nature, stochastic (Pähtz et al., 2020). Furthermore, we determine an empirical relationship for the probability of ejection as a function of the dune area ratio $r$ (equation 5) with the transition centred on $r \approx 1/2$. This is in contrast to the work of Diniega et al. (2010), who previously used a continuum model to predict the transition at $r = 1/3$. However, it is in agreement with the prediction of $r = 1/2$ from Katsuki et al. (2005). Nonetheless, the models of both Katsuki et al. (2005) and Diniega et al. (2010) are deterministic, so they cannot reproduce our probabilistic results. We also have no reason for the difference in predicted transitional value between that found by Katsuki et al. (2005) and this study, and that found by Diniega et al. (2010). This requires further investigation.

Our numerical simulations also show that coalescence occurs in two varieties, upstream- or downstream-dominant, which, together with ejection, result in three elementary types of dune interaction in 2D. Despite their probabilistic nature, these elementary types of interaction arise in specific dune size ranges and can be recognised in 3D even in the presence of turbulence and multi-directional flow regimes. Thus, they potentially provide an efficient means of decomposing coarsening phases and associated timescales (Lü et al., 2021).

Furthermore, we compare the numerical observations with ideal experimental interaction events from the study of Jarvis et al. (2022). Morphologically, the interactions appeared very similar, as evidenced by the existence of a flat plateau that appears during upstream-dominant coalescence. Additionally, the interaction outcomes agree well with the numerically determined regime diagram. Although these comparisons are favourable, restricting our simulations to a 2D domain means we are unable to reproduce the full range of dune interaction phenomena that can be observed in 3D systems (Endo et al., 2004; Durán et al., 2005; Katsuki et al., 2005; Assis and Franklin, 2020, 2021). In particular, the model cannot capture the role of transverse recirculation and the channelisation of an oblique flow between dunes when they are laterally offset. A fully 3D fluid velocity field, and possibly a variable lee slope angle, would be required for the cellular automaton model to reproduce these behaviours and the more general phenomenology of dune interactions observed in laboratory experiments, including the collision-suppression and dune-repulsion phenomena observed by Bacik et al. (2020). Our results highlight a need for experiments on interactions between two discrete dunes a) to verify if the outcomes are indeed probabilistic and b) to quantify the transition between the different regimes. Such experiments would also enable further quantitative comparison with numerical simulations, including constraints on mass exchange and velocity evolution with time.

## Appendix A:  Single-dune simulations

To determine the required domain size and initial dune separation, simulations of a single dune were performed. These sim-
ulations were initiated in a periodic domain of variable length $L$ = 600 - 19200 $l_0$ with a triangular pile of sand of variable
area (400 - 4000) $l_0^2$ and a fixed slope angle of $35°$ in the centre. During each simulation, the mound morphed into a steady
shape with a curved stoss slope and straight slip face at the angle of repose. Regardless of the size of the initial mound, this
equilibrium shape was attained within 5000 time steps (Fig. A1a) and a migration distance of $350 \, l_0$, consistent with models
for 3D barchans (Zhang et al., 2014). Hence, an initial separation $> 1000 l_0$ in the simulation is sufficient. Once equilibrated,
the position of the dune crest with time was used to calculate the migration velocity $c$. Our results for the dependency of $c$ on $A$
agree with Zhang et al. (2010), although for $L \lesssim 1200 l_0$, $c$ depends on $L$ due to long range interactions between the dune and
its repeated images in the periodic domain (Fig. A2). We are therefore justified in our choice of $L = 2000 l_0$. We also note that
the dune aspect ratio $\mathcal{H}/\mathcal{L}$ is independent of $A$ over the full range of dune sizes we used, thus demonstrating scale invariance.

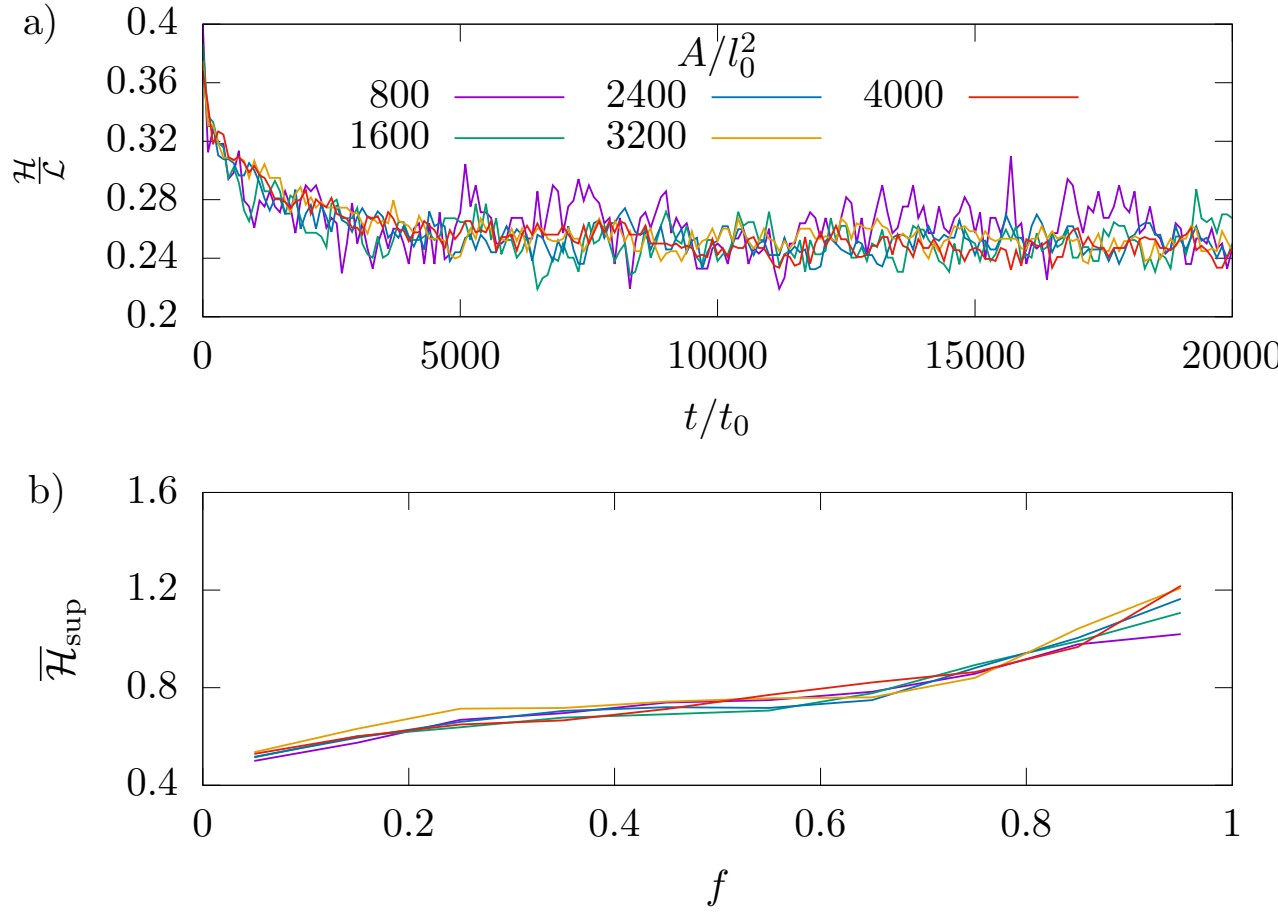

**Figure A1.** a) Time evolution of the dune aspect ratio $\mathcal{H}/\mathcal{L}$ for different dune areas $A$. The bedform is initiated as an isoceles triangle with slope angle $35°$ and an equilibrium dune shape is seen to form within $5000\ t_0$ for all areas considered. b) Time-averaged RMS amplitude of superimposed bedforms $\overline{\mathcal{H}}_{\mathrm{sup}}$ as a function of the fractional distance along the stoss slope $f$ for equilibrated dunes. It can be seen that the superimpsoed bedforms grew as they climbed the stoss slope.

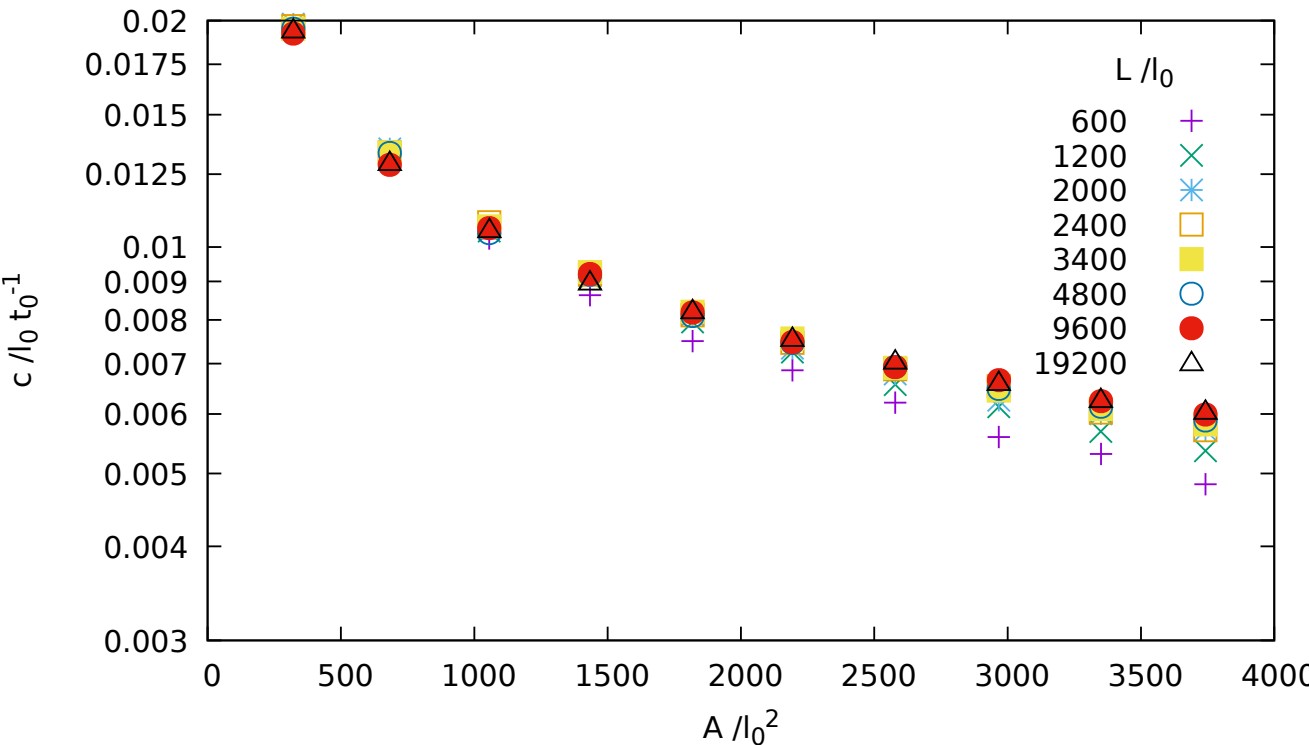

**Figure A2.** Migration velocity $c$ of an individual dune as a function of dune area $A$ and domain length $L$. We see that the dune velocity decreases with dune area. For $L \lesssim 2000 l_0$, $c$ also increases with $L$, particularly at large $A$. This is due to long range interactions between the dune and its repeated images in the periodic domain.

Superimposed bedforms were observed on the stoss slopes of these dunes (Fig. A1b). These disturbances to the dune surface
appeared at the upstream foot of the dune and propagated to the crest, growing as they climbed. Once a superimposed peak approached the dune top, it would sometimes be higher than the crest. In this case, the crest would shrink in height and a short plateau formed at the top of the slip face. The new peak migrated across this plateau whilst migration of the dune itself paused until the peak reached the slip slope and avalanching recommenced.

## Appendix B: Parameter values

Table B1 shows values of parameters used in the simulation.

**Table B1.** Values of parameters used in the cellular automaton in model and physical units. Conversion to physical units was performed using the scales determined in Zhang et al. (2014).

| Symbol | Description | Model value | Physical value |
|--------|-------------|-------------|----------------|
| $l_0$ | Length scale (i.e. cell size) | - | $3.25 \times 10^{-3}$ m |
| $t_0$ | Time scale | - | $9.74 \times 10^{-3}$ s |
| $H$ | Domain height | $300\, l_0$ | 0.681 m |
| $L$ | Domain length | $2000\, l_0$ | 4.54 m |
| $\Lambda_0$ | Transition rate for erosion | $4\, t_0^{-1}$ | 410.6 s$^{-1}$ |
| $\Lambda_c$ | Transition rate for deposition | $2\, t_0^{-1}$ | 205.3 s$^{-1}$ |
| $\Lambda_t$ | Transition rate for transport | $6\, t_0^{-1}$ | 616.0 s$^{-1}$ |
| $\Lambda_g$ | Transition rate for gravitational settling | $1000\, t_0^{-1}$ | $1.02 \times 10^5$ s$^{-1}$ |
| $\Lambda_d$ | Transition rate for diffusion | $0.02\, t_0^{-1}$ | 2.05 s$^{-1}$ |
| $\Lambda_a$ | Transition rate for avalanching | $10\, t_0^{-1}$ | $1.03 \times 10^3$ s$^{-1}$ |
| $\theta$ | Angle of repose | $35°$ | $35°$ |
| $\tau_1$ | Critical shear stress for motion | 0 | - |
| $\tau_2$ | Adjustable parameter | $1000\, \tau_0$ | - |

## Appendix C: Separation Plot

An alternative means of comparing the experimental data with the probability model (equation 5) determined from the numerical simulations is presented in Figure C1. This shows a separation plot, a visual tool that can be used to test the reliability of models with binary outcomes (Greenhill et al., 2011) To create this plot, the 19 experimental data points shown in Figure 4c are ordered from smallest $H_u/H_d$ on the left to largest on the right. Events which result in ejection are represented as red columns whilst those represented as coalescence remain white. If the model given by equation 5 were to be entirely uncorrelated with the experimental results, we would expect the red columns to be uniformly dispersed along the axis. However, there is an increasing concentration of red columns towards the right hand side of the plot, showing that ejection events indeed occur more frequently when equation 5 predicts them to be more likely. Overlain on the plot is a line representing the calculated probability of ejection given by equation 5 for each observation. This allows us to see that the only true outlier is the ejection event that occurs for $H_u = 26.0$ mm, $H_d = 46.3$ mm and has a probability of 0.002. As suggested by Figure 4c, it seems that equation 5 does a reasonably good, but not perfect, job of describing the data.

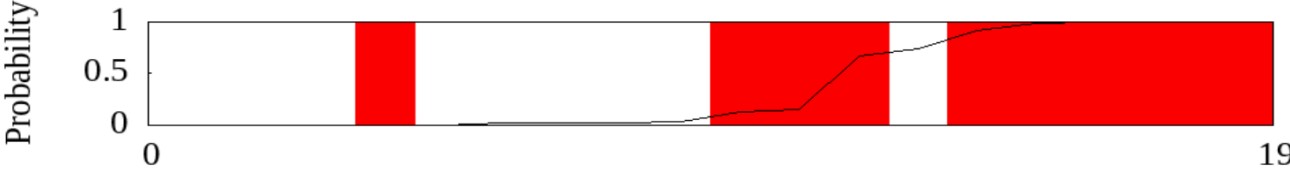

**Figure C1.** d) A separation plot, generated by ordering the 19 experimental data points from Figure 4c from smallest $H_\mathrm{u}/H_\mathrm{d}$ on the left to the largest on the right. Instances resulting in ejection are represented as red columns. Overlain is a line chart showing the probability of each interaction resulting in ejection, as given by equation 5.

*Code availability.* The ReSCAL software used to generate the results in this manuscript can be downloaded from http://www.ipgp.fr/~rozier/rescal/rescal.html

*Author contributions.* NMV conceived the study. CN and OR created the ReSCAL program. PAJ performed the simulations and data analysis and drafted the manuscript. All authors edited and contributed to the manuscript whilst NMV and CN supervised all aspects of the study.

*Competing interests.* The authors declare that they have no conflict of interest.

*Acknowledgements.* This research was funded by Royal Society Challenge Grant CH160065 and Isaac Newton Trust Early Career Grant RG 74916. NMV was supported by a Dorothy Hodgkin Fellowship DH120121 and a Royal Society University Research Fellowship No. URF/R1/191332. Some of the simulations were performed at the University of Geneva on the "Baobab" HPC cluster. The authors thank Karol Bacik and Yannik Behr for useful discussions. We thank Dominic Robson and an anonymous reviewer for insightful reviews on this manuscript, as well as Suleyman Naqshband and three anonymous reviewers for their helpful and insightful reviews and comments on an earlier manuscript. CN and OR acknowledge support from the UnivEarthS LabEx program of Sorbonne Paris Cité (ANR-10-LABX-0023 and ANR-11-IDEX-0005-002) and the French National Research Agency (ANR-17-CE01-0014/SONO).

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
