# Peer review of "The probabilistic nature of dune collisions in 2D"

_Earth Surface Dynamics, 2022_

## Author Comment (AC1)

**Response to reviewers**

We thank the reviewer for their constructive criticisms and suggestions. We have taken these on board to improve the manuscript. We hope the paper is now acceptable to both reviewers.

In the following, we respond to each of the reviewer's comments in turn. Reviewer's comments are in Italic font, with our response both indented and in Roman font.

*Overall, this work represents an interesting development in the field of Earth Surface dynamics and, in particular, the study of bedform interactions. The finding that dune interactions are probabilistic represents a novel result and may come as something of a surprise to many researchers in the community. I welcome this finding and believe that this work represents a sufficient advancement to warrant publication. The authors should also be commended for the style of the report which is generally well written, organised, and presented.*

*I do, however, have some general problems with the study. These issues centre primarily around the limitations of 2-dimensional regimes and the question of how well these results scale to realistic 3-dimensional systems. More specifically, I find the attention paid to exactly determining the form of the observed stochasticity rather irrelevant since any real-world system will behave differently because of the increased dimensionality. Additionally, the comparison with the experiments presented in Jarvis et al. (2022) is perhaps an odd choice since, as the authors point out, those experiments involved a "train" of interacting dunes rather than the binary system considered in the simulations presented in this work. Although the authors mitigate some of the issues this might present by considering only interactions where there was no physical overlap with the additional bedforms, there may be wake effects such as those identified in Bacik et al. (2020) which are not accounted for in this work. I believe that the discussion of the limitations of this work should be made more detailed as, in my opinion, these problems are more substantial than they are made to seem in the current manuscript. Nevertheless, the essential finding that collisions are stochastic rather than deterministic is an important one and the authors should be congratulated for their work.*

> We thank the reviewer for this comment. We agree that there are limitations to this work (as there is to all studies) and have made an effort to be more detailed in this, as described in our response to specific comments below. Regarding the specific wake effects described by Bacik et al. (2020), the author is correct that these are not accounted for in the numerical model. Indeed, they cannot be reproduced due to the pure 2D geometry of the numerical domain. We now try to better acknowledge this throughout the paper (see our responses to specific comments below for exact line numbers).

*I will now provide some more specific comments.*

*Specific Comments:*

*Line 25 - "pattern coarsening, whereby a larger number of smaller dunes transition to become a smaller number of larger dunes…"*

*Coarsening has been observed in many experimental and numerical studies of dune dynamics. However, many natural dune systems have been shown to be homogeneous rather than coarsening. This point should be explicitly made here.*

We have now added the line "This coarsening process occurs during the development of a dune field from a flat bed whereas more mature dune fields can attain a steady state, where their mean wavelength and amplitude remain relatively constant." Lines 28-30.

Line 41-42 -*"Turbulence, however, is an inherently 3D phenomenon…"*

*Interesting claim given that the water tank experiments of Bacik et al. (2020) reported that induced turbulence led to repulsion even in their quasi-2D setup. Furthermore, the claim that only the size ratio controls the collision outcome is likely only true for sufficiently large bedforms where the assumption of scale invariance applies.*

We now clarify that the size ratio is the only controlling parameter if the dunes are sufficiently large to be scale invariant (Lines 46-47). We also explicitly state that the dunes in our simulations are scale invariant (Lines 291-292). Regarding the turbulence, we note that the quasi-2D experiments of Bacik et al. (2020) are inherently 3D. Thus, turbulence can be, and is, present in those experiments, as we now note on Line 45. Our simulations, however, are pure 2D. Consequently, vortices, such as the recirculation zone behind the crest of dunes, cannot decay into a 3D turbulent field.

Line 59 - *"However this means we can exhaustively…"*

*The merit of such an exhaustive study is severely limited however by the fact that 3D and 2D systems are inherently different. Although you may be able to fully understand the problem in 2D, one must recognise that this is still a toy model and that a study in 3D is going to have more real-world impact even if it cannot be quite as exhaustive.*

We fully agree that we are not capturing phenomena that will be present in a 3D system. However, 3D systems contain so much complexity, the fundamental result which we observe here, namely the fact dune collisions can be modelled probabilistically, would be much harder to identify in the wider parameter space. To address this, we have added the sentence "Although our study is strictly only valid for 2D systems, our results should motivate further research to test if the fundamental characteristics of collisions that we observe can be preserved in 3D environments, where turbulence and flow perturbations also have an influence." (Lines 69-71)

Line 60 -*"By performing large numbers…"*

*This makes it sound as though thousands of simulations have been performed but we later find out that it was only ~50 and that, in fact, the number of simulations that could be performed was a limiting factor on the uncertainty of the findings.*

To be clear, we have performed many more than 50 simulations. In fact, each data point in Fig. 3D represents 50 simulations alone. Fig. 3D contains 32 data points. Thus, we actually performed 32 × 50 = 1600 simulations. We now state clearly that we performed a large number of simulations (line 65) as well as re-emphasise this in the results (line 198).

Line 126 -*"Two distinct types of coalescence…"*

*This phrasing makes it sound like they are very different processes, however the authors themselves describe that the intermediate stages are only "slightly different" and later (line 151) state explicitly that "...close to the transition it is very difficult to distinguish the two types of behaviour". This suggests that the types of coalescence are not really "distinct" as claimed here but two regimes of a single coalescence process between which there exists a continuous transition.*

Our results show that the two types of coalescence, upstream-dominant and downstream-dominant, are distinct. This is most clearly demonstrated in Fig. 1. It can be shown that, for downstream-dominant coalescence, the sediment from the upstream dune (red) is preserved in a slip slope-parallel layer in the final dune whereas, for upstream-dominant coalescence, the red sediment remains preserved in a mixed region above an unmixed basal layer preserved from the downstream dune (blue). These very distinct morphologies clearly demonstrate that the two types of coalescence are distinct processes and not end-members of a single phenomenon. We have now reworded the manuscript to make this clear (lines 153 and 189).

*Line 149 - "...so many simulations would be required to gain meaningful outcomes."*

*But the authors claimed in the introduction that they had performed "large numbers" of simulations and were able to "exhaustively study" the phenomena. This is a direct contradiction and makes it seem as though the claims made in the introduction were unwarranted.*

The line in question refers specifically to the transition between downstream- and upstream-dominant coalescence. The key focus of our manuscript is to quantify the transition between coalescence and ejection. We have now edited the manuscript to make this clearer (line 60).

*Line 160 - "...creating multiple small bedforms"*

*Would it not make sense to define the cases where different numbers of bedforms were generated as different types of collisions, particularly as in 3D it may be possible for these new bedforms to escape from between the dunes? This would also be more consistent with the distinction between the types of coalescence identified by the authors for which the intermediary stages were key.*

We prefer to maintain the current classification, whereby the events which result in more than two bedforms being created are classified as ejection. This is because, ultimately, the underlying mechanism is the same regardless of whether one or more downstream dunes are ejected. However, in order to quantify the effect of initial size ratio on the number of dunes created would require many more simulations than we currently perform, since there would be many more possible outcomes. We have now added some text to expand on this (lines 170-173).

*Line 188 - "Performing further simulations…"*

*Again, this ought to be mentioned earlier as the introduction makes it seem that the study was not restricted in this manner.*

Please see our above responses to your comments on the number of simulations, and our focus on quantifying the coalescence-ejection transition

*Line 194 - "finding a = 14±2 and b = 0.509±0.005."*

*Given that the results of 2D experiments are not likely to be fully scalable to 3D I do not believe that exact determinations of these constants are particularly relevant to real-world systems. As such, I think these values could easily be removed to an appendix.*

We agree that these values are specific to the 2D case and, given the larger dimensionality of the parameter space and the greater number of possible outcomes, are not directly applicable to 3D systems. However, these results are a key result for the 2D system we study here. Additionally, we rely on these values to perfom the comparison to experiments in Section 4. We therefore chose to include them. However, we now acknowledge in the manuscript that these fitted values are not appropriate for 3D systems (lines 208-209).

*Figure 3 caption -"...is due to only 50 simulations…"*

*Same point made previously, this is a major shift in tone from the introduction.*

> Please see our responses to the above comments. In particular, we note that each data point in Figure 3d corresponds to 50 simulations and we have actually performed 1600 simulations in total.

*Line 204 -"...only simulated interactions between two discrete dunes… train of interacting dunes"*

*I think this is a more important caveat than the authors make it seem. Other similar studies (e.g. Bacik et al. (2020) have found that in these systems wake induced turbulence plays a critical role. The turbulence generated by multiple interacting bedforms in these experiments is likely to be greatly affecting the outcomes. Whereas, this is not the case in the simulations where only two dunes were present.*

> We agree with the reviewer that this is an important caveat and we do not wish to underplay this in the manuscript. As we already state in the manuscript (lines 226-228), we attempt to mitigate this by selecting interactions a) between discrete dunes and b) during times when there was no physical overlap with neighbouring dunes. However, we now explicitly acknowledge that the presence of a dune train will lead to fluid-transmitted interactions which we cannot remove (lines 228-229).

*Line 205 -"...the number of simulations… is relatively small"*

*Same point made previously!*

> Please see our our above responses on this point.

*Line 209 -"However, additional simulations…"*

*If these simulations have been performed already and the authors wish to compare with the experimental results of Jarvis et al. (2022) then why not simply present the results from the experiments where θ = 18â¦ rather than those where θ = 35â¦ ?*

> The text in the previous version of the manuscript was erroneous and misleading. Although we have performed some simulations with a leeside slope angle of $θ = 18°$, they are only indicative, showing that ejection and coalescence occur as observed in the simulations where $θ = 35°$. We have now clarified in the manuscript this caveat (lines 224-226).

*Line 225 -"There is reasonable agreement…"*

*I would like to see a table containing these data and ideally some statistical tests about whether the experimental observations are statistically similar to the stochastic rules defined in this work.*

> Following Greenhill et al. (2011) and Davidson-Pilon (2015) (https://nbviewer.org/github/CamDavidsonPilon/Probabilistic-Programming-and-Bayesian-Methods-for-Hackers/blob/master/Chapter2_MorePyMC/Ch2_MorePyMC_PyMC3.ipynb) we now present a separation plot comparing the predicted and observed number of ejection events. We chose to use a visual comparison  rather than a quantitative statistical test since, such tests often rely on arbitrary thresholds, e.g., the expected Percentage of Correct Predictions (Heron, 1999), or lack statistical interpretations, e.g., Brier Scores (Brier, 1950).

As can be seen in our new Fig. 4D, and as described in the text (lines XX – XX), the separation plot shows that the model suggested in equation 5 does a reasonably good job of predicting the experimental data, since the majority of ejection events occur on the right hand side of the plot.

Finally, as suggested by the reviewer, we have now inlcuded a table (Table 1) detailing the dune collisions selected from the experiments and the modelled probability of ejection for each event.

*Line 253 -"...agree well with numerical determined…"*

*Again I would like to see these data.*

Please see our response to the above comment.

**References (if not in paper)**

Brier, G. W. (1950). Verification of Forecasts Expressed in Terms of Probabilities. *Bull. Am. Meteorol. Soc.*, 78, 1-3.

Herron, M. C. (1999). Postestimation Uncertainty in Limited Dependent Variable Models. *Polit. Anal.*, 8(1), 83-98.

---

## Author Comment (AC2)

**Response to reviewers**

We thank the reviewer for their constructive criticisms and suggestions. We have taken these on board to improve the manuscript. We hope the paper is now acceptable to both reviewers.

In the following, we respond to each of the reviewer's comments in turn. Reviewer's comments are in Italic font, with our response both indented and in Roman font.

*The manuscript presents a numerical study of dune-dune collisions in two-dimensions. For that, the authors carried out cellular automaton simulations and compared the results with quasi-2D experiments. The subject is interesting, and the manuscript is well written and should be considered for publication. However, I have some concerns that I list below.*

**General comments**

*- You affirm that the collisional processes are not deterministic. In my opinion, you must better justify this affirmation, or reformulate some of your sentences. For me, the physics here is deterministic, since the motion of each sand grain is deterministic. Of course, one can analyze or model the problem as probabilistic, but, in principle, it is (I believe) deterministic.*

> We have rephrased some sentences to make clear that we are modelling the process as probabilistics (lines 66-67, 193-194, 263-264)

*- After briefly discussing the turbulent wake shed by the upstream dune (line 39), you state that details of turbulence are negligible in the 2D simulations because turbulence is inherently 3D. However, the presence of a recirculation bubble in the wake of the upstream dune (independent of turbulence, since it can simply be a recirculation region) can affect significantly the dune-dune collision (even avoiding it, as shown in the experiments of Bacik et al., PRL, 2020). In addition, 2D dunes in nature (or in labs) have a finite thickness, and, therefore, the flow can be turbulent. Please consider reformulating your sentences.*

> We agree with the reviewer that, even in 2D systems, the recirculation region downstream of a dune's crest can affect dune-dune collisions. Critically however, in 3D systems, this wake will decay into a 3D turbulent field. This is the case in the experiments of Bacik et al. (2020) which, although they take place in a narrow flume, are now pure 2D. In our pure 2D simulations, however, the wake behind a dune cannot produce turbulent fluctuations. We have now tried to explain this better in the revised manuscript. For line numbers of specific changes, please see the response to specific comments below.

*- You compare your numerical results against those of Jarvis et al. J. Geophys. Res: ES, 2022, in which a train of dunes was present. Please consider comparing your results also with the experiments of Bacik et al., PRL, 2020. For example: can your simulations reproduce the dune-dune repulsion observed by Bacik et al.? If not, why?*

> The simulations cannot reproduce the dune-dune repulsion observed by Bacik et al. (2020). This is because the repulsion phenomena occurs through the action of turbulent wakes downstream of dune crests. Since we are in a pure 2D system, unlike the quasi-2D system of Bacik et al. (2020), the simulations do not produce turbulent wakes and thus dune-dune repulsion does not occur. We have now edited the introduction of the manuscript to try and explain better why we do not observe dune-dune repulsion in these simulations (lines XX).

*- In my opinion, in their current form the comparisons with experiments are most qualitative. In order to be more quantitative, you should present the dune profiles (perhaps superposed), celerity of crests, values of mass exchanges, etc. This would strengthen your conclusions.*

> The focus of this manuscript is to specifically consider the transition between coalescence and ejection, rather than the general properties of collisions. Therefore, we feel that Fig. 4C presents a quantitative comparison between simulations and experiments for this purpose. Whilst comparing other properties may be interesting, e.g., migration speed, mass exchange values, these wouldn't help to constrain the transition and, thus, we don't consider them in this study.

**Specific comments**

*- The manuscript is well written and agreeable to read, congratulations for that.*

*- Line 24, "Such collisions have been frequently observed in subaqueous experiments and numerical simulations…". They have also been observed, although with incomplete time series, for aeolian dunes.*

> We have now edited lines 25-26 to include this.

*- Lines 27-32: It should be stated clearly here that these collisional processes are specific for 2D dunes. For 3D dunes, processes are more complex, with the existence of splitting mechanisms and other types of ejection (of new dunes), as you explain briefly in the next paragraph. In addition, it is not only the dune-dune collision that redistributes sand, but also dune-dune interactions through disturbances of the fluid flow (mainly the wake of the upstream dune), which can cause, for instance, surface waves (on the downstream dune) and, consequently, calving.*

> We now clearly state that coalescence and ejection are specific to 2D dunes (line 39) and now also refer to wake interactions causing calving (lines 42-44)

*- Lines 41-42: the presence of a recirculation bubble in the wake of the upstream dune (independent of turbulence, since it can be simply a recirculation region) can affect significantly the dune-dune collision (even avoiding it, as shown in the experiments of Bacik et al., PRL, 2020). In addition, 2D dunes in nature (or in labs) have a finite thickness, and, therefore, the flow can be turbulent. Please consider reformulating your sentence.*

> We agree with the reviewer that recirculation, even in the absence of turbulence, can affect the collision. However, for dunes sufficiently large to be self-similar, the recirculation zone length scales with the dune size. Thus, in pure 2D, the collision outcome still only depends on the size ratio between the dunes. We have reworded this section to make this clear, as well as emphasise that 3D turbulence will be present even in quasi-2D laboratory experiments (lines 45 – 47).

*- Line 51: Please consider stressing here that those results are, in principle, valid only for subaqueous barchans, and have not been tested against aeolian barchans.*

> We have now stated that the results in Assis & Franklin (2020) are for subaqueous barchans (lines 55-57)

*- Line 58: Ok. But please note that 2D dunes in laboratories have finite thicknesses and are not strictly 2D.*

As stated in the response to the above comment, we now emphasise that quasi-2D dunes in experiments are, in reality, 3D (lines 45-46). However, we wish to remphasise that the numerical simulations are pure 2D.

*- Line 64, about comparison with experiments: Please consider comparing your results also with the experiments of Bacik et al., PRL, 2020.*

As stated in response to an above comment, our numerical results are not directly comparable to the results of Bacik et al. (2020). The quasi-2D nature of the experiments in Bacik et al. (2020) means that turbulent wakes occur, and these cause dune-dune repulsion phenomena. In our pure 2D cellular automaton, these turbulent wakes are not reproduced (although recirculation zones are) and, therefore, we cannot obtain dune repulsion. We have reworded the introduction to try and make clear that effects caused by 3D turbulence, including dune repulsion, cannot be reproduced with this model.

*- Lines 75-76: On the one hand, experiments and DNS show that the slope angle of the leeside is important, and, on the other hand, your results are based on a probabilistic approach/analysis: should not the model consider the slope angle as a (stochastic) variable?*

For a single simulation within the cellular automaton model, it is necessary to set the angle of repose, and therefore the maximum slope angle, as a constant input to the model. While it would be possible to run lots of simulations with different angles of repose drawn from a probability distribution (effectively treating it as a stochastic variable) this would present both practical and scientific difficulties. Practially, this would require substantially increasing the number of simulations beyond the 1600 we have already performed. Scientifically, randomly selection values of the angle of repose from a probability distribution prevents any accountability of the physical controls on this parameter.

We therefore chose to not explicitly consider the effect of angle of repose on the collision dynamics in this study.

*- Lines 76-77: I find this sentence strange: for me, the physics here is deterministic. One can analyze or model the problem as probabilistic, but, in principle, the motion of each sand grain is deterministic.*

The cellular automaton model is probabilistic, not deterministic. Importantly, the model does not solve for the motion of individual sand grains. Instead, as described in lines 77 – 85, sediment transport is modelled using transitions of pairs of nearest-neighbour cells corresponding to physical processes (erosion, deposition, transport). The rates for these transitions are set as input parameters and determine the probability for a particular transition to occur. Precise details on the model can be found in Narteau et al. (2009) and Rozier and Narteau (2013).

*- Lines 95-96: I do not totally agree. I would expect variations in quasi-2D experiments (or simulations), since the flow disturbances (wake) generated by the upstream dune vary with the flow strength.*

In 3D (or quasi-2D) simulations, we agree that flow strength would impact the wake downstream of dunes. In our pure 2D simulations, where the flow regime is also far from the sediment transport thresold, the length of the recirculation zone is determined purely by the size of the dune. We have now tried to better explain this in the manuscript.

*- Line 100, about comparison with experiments: Again, please consider comparing your results also with the experiments of Bacik et al., PRL, 2020.*

Please see our comment above where we have addressed this.

*- Lines 135-136: Please note that you compare your data with dunes that are not strictly 2D (then, turbulence may be important)…*

We now explicitly stated throughout the manuscript, most notably in lines 221 – 224.

*- Lines 163-179. This part deserves a deeper discussion: you could present some statistics of mass exchange in order to strengthen your points. Another thing is that you should better justify your analysis, since your simulations do not compute the trajectory of each grain based on Newton's second law: as a reader I would expect a discussion on how accurate the simulations are and if they are physically consistent.*

> The paragraph the reviewer refers to qualitatively describes the morphological structures that are created during the dune collision, as depicted in Figure 1. This figure clearly shows that the morphology of each type of interaction is clearly distinct, even without quantification. We agree with the reviewer that aspects of this could be quantified, e.g., the proportion of mass which is exchanged during the interaction. However, the aim of this paper, is to describe and quantify the transition between coalescence and ejection. We are not aiming to go into quantitatively describing the different types of transition themselves.

> On the $2^{nd}$ point, the reviewer is correct that the simulations do not explicitly compute sand grain trajectories. However, the model has previously been shown to reproduce dune dynamics successfully. The model produces dunes from unidirectional flow over a flat bed (Narteau et al., 2009) and this emergent behaviour can be compared with natural systems to set the length and time scales of the model (Narteau at al., 2009; Zhang et al., 2010; 2014). Beyond this, the model has also been shown to predict pattern coarsening (Gao et al., 2015).

*- Lines 182-183: This affirmation is rather strong. In my opinion, the processes are deterministic (since each grain follows Newton's second law). What happens is that one can analyze the problem (from experiments, for example) from a probabilistic point of view, or use a probabilistic model to compute/simulate the processes. The fact that we can use a probabilistic model that works does not mean that the problem is not deterministic in essence.*

We have now reworded this sentence to clarify this (lines 194-195).

*- Line 199: I do not agree. In my opinion, the comparisons are most qualitative. In order to be more quantitative, you should present the dune profiles (perhaps superposed), celerity of crests, values of mass exchanges, etc.*

> We have now reworded the sentence to be clear that we are comparing the morphology of the interacting beforms qualitatively and the regime transition quantitatively (lines 212-213). Whilst all of the other suggestions are possible, as noted above, the primary aim of this paper is to quantify the regime transition. The additional proposed analysis would not contribrite towards this.

*- Lines 210-211: You should better justify this assertion.*

The text in the previous version of the manuscript was erroneous and misleading. Although we have performed some simulations with a leeside slope angle of $\theta = 18$ º, they are only indicative, showing that ejection and coalescence occur as observed in the simulations where $\theta = 35$ º. We have now clarified in the manuscript this caveat (lines 225-227).

*- Lines 214-215: So, you could not compare the results for this case. Can you find published data of experiments for this case?*

Experimental observations of downstream-dominant coalescence have been reported by Coleman & Melville (1994). However, these were between continuous, transverse bedforms. To the authors knowledge, there are no experimental data concerning downstream-dominant coalescence between discrete dunes.

*- Lines 236-237. Again: you need to better justify this affirmation. In my opinion, the process is not probabilistic in essence (it is deterministic), but can, perhaps, be analyzed (or computed) in a probabilistic way.*

We have reworded this sentence to make this clearer (lines 264-265).

---

## Referee Report (RR1)

General comments:

The authors have taken on board many of the suggestions from myself and the other reviewer and the changes they have made to address our comments have improved the manuscript. In particular, I felt that the limitations of the study were not made clear enough in the original manuscript whereas, in the revised manuscript, the authors have made an effort to highlight these limitations to a greater extent. I am still of the opinion that precise quantitative analysis of two-dimensional simulations (and experiments) has very little relevance to real-world dunes as acknowledged by the authors, which limits the impact of studies such as this. However, there has been a precedent in recent years for the publication of similar studies and so this work does merit publication and I would like to again commend the authors for their work.

Specific comments:

Line 70 - "Although our study is strictly only valid…"

- New sentences like this that make the limitations more explicit have greatly improved the manuscript.
- It is also important to note though, that the differences between 2D and 3D systems are not solely confined to properties of the flow but also to the interactions (e.g. avalanching etc.) between longitudinal cross-sections in a dune.

Line 210 - "It is important to note…"

- It's good that you have made this more explicit.

Line 212 - "We compare…"

- Perhaps the authors could explain here why the repulsion observed in Bacik et al. (2020) was not observed in Jarvis et al. (2022). The authors have sufficiently explained why their model cannot reproduce the wake-induced repulsion but have not explained why that effect was not observed in their chosen quasi-2D experiments.

Figure 4 and lines 251-261

- I am not sure if the separation plot is the easiest figure to interpret although it was helped by the paragraph the authors included. I did not feel that this added to the work.

---

## Author Response (AR2)

**Response to reviewers**

We thank both reviewers for their further constructive criticisms and suggestions. We have taken these on board to improve the manuscript. We hope the paper is now acceptable to both reviewers.

In the following, we respond to each reviewer's comments in turn. Reviewer's comments are in Italic font with our response both indented and in Roman font.

Along with these responses, we have uploaded two copies of the revised manuscript, a marked-up version showing changes from the previous submission as well as a "clean" copy. Where these responses include line numbers, these refer to the marked-up version.

Reviewer 1

*General comments:*
*The authors have taken on board many of the suggestions from myself and the other reviewer and the changes they have made to address our comments have improved the manuscript. In particular, I felt that the limitations of the study were not made clear enough in the original manuscript whereas, in the revised manuscript, the authors have made an effort to highlight these limitations to a greater extent. I am still of the opinion that precise quantitative analysis of two-dimensional simulations (and experiments) has very little relevance to real-world dunes as acknowledged by the authors, which limits the impact of studies such as this. However, there has been a precedent in recent years for the publication of similar studies and so this work does merit publication and I would like to again commend the authors for their work.*

> We thank the reviewer for their positive comments and their honest appraisal of this work, during both rounds of the review process so far. We acknowledge that the extent to which 2D experiments and simulations are relevant to dunes in the natural world remains uncertain, and that there is clearly a gap between the complexity of the processes we consider here and what occurs in nature. Hopefully, this gap can be addressed as new knowledge is gained, both through studies which consider detailed specific subsets of processes as well as those which consider a more comprehensive viewpoint.

*Specific comments:*
*Line 70 - "Although our study is strictly only valid…"*
> *● New sentences like this that make the limitations more explicit have greatly improved the manuscript.*
> *● It is also important to note though, that the differences between 2D and 3D systems are not solely confined to properties of the flow but also to the interactions (e.g. avalanching etc.) between longitudinal cross-sections in a dune.*

> > We have now added an extra clause to this sentence to make clear that lateral sediment transport during collisions is also an important difference between 2D and 3D systems (lines 73-74).

*Line 210 - "It is important to note…"*
  ● *It's good that you have made this more explicit.*
*Line 212 - "We compare…"*
  ● *Perhaps the authors could explain here why the repulsion observed in Bacik et al. (2020) was not observed in Jarvis et al. (2022). The authors have sufficiently explained why their model cannot reproduce the wake-induced repulsion but have not explained why that effect was not observed in their chosen quasi-2D experiments.*

  There are multiple possible reasons why repulsion was not observed in Jarvis et al. (2022). In particular, it is important to note that, in the experiments of Bacik et al. (2020), which considered the interaction between a pair of dunes in a periodic domain, dune repulsion acted to push the system to a state where the dunes would find equilibrium in antipodal positions. In the experiments of Jarvis et al. (2022), the dunes were allowed to form spontaneously from a flat bed. Consequently, at the time of interactions between discrete dunes, there were approximately 10 dunes, almost evenly spaced around the flume circumference. Since these dunes were already almost evenly spaced, if dune repulsion did occur, the effects would have been very small and difficult to observe. We have now added text to the manuscript to explain this (lines 236-241).

*Figure 4 and lines 251-261*
  ● *I am not sure if the separation plot is the easiest figure to interpret although it was helped by the paragraph the authors included. I did not feel that this added to the work.*

  We have moved the separation plot and the associated text to a new Appendix C (lines 320-331 and Fig. C1).

Reviewer 2

*Dear authors,*

*You revised the manuscript and answered to my comments properly. Although in many parts I still disagree with you (you will see from my comments below), I recognize that this is a well conducted work that represents a significant contribution to the field. Therefore, my advice is the manuscript acceptance.*

*Comments*

*- You could have sent a marked-up version. This would facilitate the task of reviewers. Please consider doing that next time.*

  We did upload a marked-up version and apologise to the reviewer if this was not clear. We have ensured to do this again.

*- I am satisfied with the modified sentences on the probabilistic nature of the method (opposed to the deterministic characteristics of the problem itself).*

> We are pleased that we were able to address this to your satisfaction and feel that the manuscript is better for this.

*- I reproduce here one of my comments in the previous review: "After briefly discussing the turbulent wake shed by the upstream dune (line 39), you state that details of turbulence are negligible in the 2D simulations because turbulence is inherently 3D. However, the presence of a recirculation bubble in the wake of the upstream dune (independent of turbulence, since it can simply be a recirculation region) can significantly affect the dune-dune collision (even avoiding it, as shown in the experiments of Bacik et al., PRL, 2020). In addition, 2D dunes in nature (or in labs) have a finite thickness, and, therefore, the flow can be turbulent. Please consider reformulating your sentences".*

*I do not totally agree with your answer. Turbulence can be inherently 3D, but not the recirculation region, which can exist in 2D (or quasi-2D) flows. Under high confinement, one should expect strong effects of this recirculation region on dune-dune collisions, as shown by Bacik et al., PRL, 2020. This is strongly related with another of my comments in the previous review: the simulations should reproduce the results of Bacik et al., PRL, 2020. Otherwise, it seems that there is something to be fixed.*

> We fully agree that the recirculation zone can exist in 2D flows. Indeed, it is present in our simulations which are performed in a pure 2D domain, with flow separation at the dune crest and reattachment at some point downstream. However, the presence of a recirculation zone alone does not lead to the collision-suppression and dune-repulsion phenomena reported by Bacik et al. (2020). In fact, Bacik et al. (2020) show that, in their experiments, the migration velocity of the downstream dune is strongly influenced by fluctuations in sediment transport caused by the turbulent wake shed by the upstream dune. This turbulent wake is generated by decay of vortical structures into a 3D turbulence field. Thus, in a 2D domain, this decay does not occur, and the small-wavelength fluctuations do not form. Consequently, collision-suppression is an inherently 3D phenomenon and cannot be reproduced in our simulations.

> We have added a clause to try and better explain this in the manuscript (lines 46-47 and lines 291-297).

*- I reproduce here other of my comments in the previous review: "You compare your numerical results against those of Jarvis et al. J. Geophys. Res: ES, 2022, in which a train of dunes was present. Please consider comparing your results also with the experiments of Bacik et al., PRL, 2020. For example: can your simulations reproduce the dune-dune repulsion observed by Bacik et al.? If not, why?"*

*Here again, I do not totally agree with your response: Since the experiments of Bacik et al., PRL, 2020 were conducted in a Hele-Shaw circular flume, then their results should tend to a 2D*

*problem. For instance, their results for 2D dunes are different from the barchan-barchan case, and the differences are assumed to be due to confinement. Perhaps the cause for the simulations not reproducing the results of Bacik et al., PRL, 2020 is some other limitation of the numerical method. I suggest that you consider that in future works.*

> The flume used in the experiments of Bacik et al. (2020) has a width *W* of 9 cm, whilst the total flow depth *D* is 40 cm. This is an aspect ratio of *W/D* = 0.225. Additionally, Figure 3d of Bacik et al. (2020) shows a dune height of about 8 cm. Therefore, the width of the dunes in these experiments is comparable to their height. Consequently, although the channel is sufficiently thin to reduce (albeit not remove) lateral variation in dune morphology, the spatial scale of turbulent fluctuations in the fluid flow is significantly smaller than the channel width. Therefore, whilst the mean fluid flow will tend to a 2D flow field, the turbulent fluctuations will still be 3D. We think this is a strong argument for why the 2D simulations fail to reproduce the simulations of Bacik et al. (2020).

> Nevertheless, we concur that there may be other limitations of the numerical model and that fully 3D simulations are necessary to verify this. We now acknowledge this in the manuscript (lines 291-297).

*- I still consider that some more quantitative comparisons should be incorporated.*

> Although we are unable to include further quantitative comparisons in this manuscript, we agree that such work would be highly valuable going forward. Indeed, this is something we would like to consider for further work. We have added some text to the manuscript to expand on this (lines 298-299).

*- I reproduce here other of my comments in the previous review: "On the one hand, experiments and DNS show that the slope angle of the leeside is important, and, on the other hand, your results are based on a probabilistic approach/analysis: should not the model consider the slope angle as a (stochastic) variable?"*

*I understand the additional extra work that this would engender, but I believe that the slope of the leeside is crucial for what happens to the downstream dune. Could this be one of the causes for not reproducing the results of Bacik et al., PRL, 2020?*

> We acknowledge that the leeside slope angle may very well play some role in the behaviour of the downstream dune. However, we feel that the 2D-3D difference is the more significant factor in the inability of the simulations to reproduce the dune repulsion of Bacik et al. (2020). Nonetheless, we have now added some text to the manuscript to emphasise that allowing for a variable lee slope angle may be necessary to fully capture the complexity of dune-dune interactions (lines 294-297).

*- You started your answer to one of my comments with "The cellular automaton model is probabilistic, not deterministic". I know that, and never stated the contrary. The problem with*

*the previous version of the manuscript is that the sentences were misleading (you affirmed in that version that the collisional process was not deterministic…).*

We never intended to suggest that you were unaware of how cellular automaton models work. Instead, we wanted to state everything in the clearest possible terms to try and be clear and minimise any confusion. Most importantly, we wanted to avoid assuming any knowledge. We are happy to hear that the reviewer is satisified with our sentences on the probabilistic nature of the method.